# BRIDGING ADVERSARIAL SAMPLES AND ADVERSARIAL NETWORKS

## ABSTRACT

Generative adversarial networks have achieved remarkable performance on various tasks but suffer from sensitivity to hyper-parameters, training instability, and mode collapse. We find that this is partly due to gradient given by non-robust discriminator containing uninformative adversarial noise, which can hinder generator from catching the pattern of real samples. Inspired by defense against adversarial samples, we introduce adversarial training of discriminator on real samples that does not exist in classic GANs framework to make adversarial training symmetric, which can balance min-max game and make discriminator more robust. Robust discriminator can give more informative gradient with less adversarial noise, which can stabilize training and accelerate convergence. We validate the proposed method on image generation tasks with varied network architectures quantitatively. Experiments show that FID score of generated samples is improved by $10\% - 50\%$ relative to the baseline with additional $25\%$ computation cost. Training stability is improved and mode collapse is alleviated.

## 1 INTRODUCTION

Generative adversarial networks (GANs) have been applied successfully in various research fields such as natural image modeling (Radford et al., 2015), image translation (Isola et al., 2016; Zhu et al., 2017), cross-modal image generation (Dash et al., 2017), image super-resolution (Ledig et al., 2016), semi-supervised learning (Odena, 2016) and sequential data modeling (Mogren, 2016; Yu et al., 2016). Different from explicit density estimation based models (Kingma et al., 2014; Oord et al., 2016; Hinton, 2012), GANs are implicit generative models with two neural networks playing min-max game to find a map from random noise to target distribution, in which the generator tries to generate fake samples to fool discriminator and the discriminator tries to distinguish them from real samples (Goodfellow et al., 2014). In original GANs formula, optimal discriminator measures the Jensen-Shannon divergence between real data distribution and generated distribution. The discrepancy measure can be generalized to f-divergence (Nowozin et al., 2016) or replaced by earth-mover distance (Arjovsky et al., 2017). Despite the success, GANs are notoriously difficult to train(Kodali et al., 2018; Arjovsky & Bottou, 2017), which are very sensitive to hyper-parameters. When the support of these two distributions are approximately disjoint, gradient given by discriminator with standard objective may vanish, and training becomes unstable (Arjovsky et al., 2017). More seriously, generated distribution can fail to cover the whole data distribution and collapse to a single mode in some cases (Dumoulin et al., 2017; Che et al., 2016).

The condition of discriminator determines the training stability and performance to a great extent. On the one hand, representation capacity of discriminator realized by a neural network is not infinite. Meanwhile, the discriminator is usually not optimal to measure true discrepancy when trained in an alternative manner practically. On the other hand, discriminator as a classifier is also vulnerable to adversarial samples (Appendix D): benign samples added by imperceptible perturbation can mislead classifier to give wrong prediction (Szegedy et al., 2014)(Goodfellow et al., 2015). Adversarial samples can be easily crafted by gradient-based method such as Fast Gradient Sign Method (FGSM) (Goodfellow et al., 2015) or Basic Iterative Method (BIM) (Kurakin et al., 2017). It should be noted that **the gradient given by discriminator that guides update of the generator is exactly the same as gradient used to craft adversarial samples of the discriminator**. In other words, the gradient contains uninformative adversarial noise which is imperceptible but can mislead the generator. However, generator can still generate meaningful samples in classic GANs training procedure. This

is because discriminator is adversarially trained with diverse generated fake samples. Nevertheless, adversarial training on real samples does not exist in classic training framework. As a consequence, training will become unstable when generated distribution approximates target distribution because the gradient given by non-robust discriminator around real samples contains more adversarial noise. To this end, we introduce adversarial training on real samples into classic GANs training framework to further improve the robustness of discriminator, which can reduce adversarial noise contained in gradient. It can be proved by results shown in Figure 1 empirically that the noise in gradient of adversarially trained discriminator is partly eliminated. Meanwhile, our proposed method can regularize the capability discriminator by performing adversarial training both on real samples and fake samples to alleviate training collapse. We validate the proposed method on image generation tasks with widely adopted DCGAN (Radford et al., 2015) and ResNet (He et al., 2015; Gulrajani et al., 2017) architecture, which shows consistent improvement of training stability and acceleration of convergence. To our best knowledge, this is the first work to consider GANs from the perspective of adversarial samples, besides which we make GANs training scheme symmetric and improve performance efficiently with acceptable computation cost. We term the proposed method as adversarial symmetric GAN (AS-GAN).

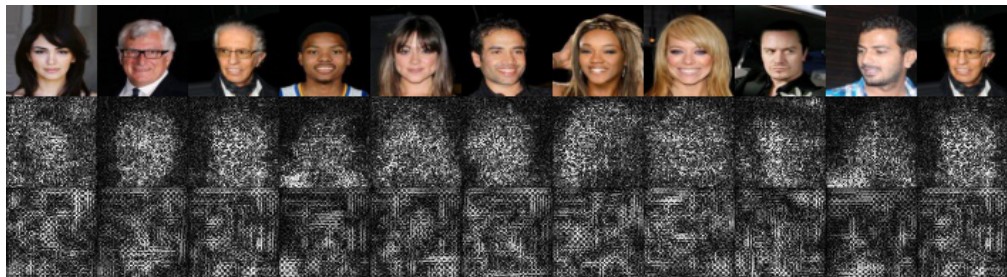

Figure 1: Visualization of the gradient of DCGAN discriminator with respect to input images. The first row shows samples from CelebA dataset. The second row and third row show gradients of adversarially symmetrically trained discriminator and standard discriminator, respectively. We clip gradient to within $\pm 3$ standard deviations of their mean and take the average absolute value of three channels for easy visualization. We can see that adversarially trained discriminator can provide more informative gradient with less adversarial noise pattern, which can stabilize GAN training.

## 2    RELATED WORK

There is a large body of work on how to stabilize GANs training and alleviate mode collapse. (Arjovsky & Bottou, 2017) proved that the widely adopted non-saturating loss function for the generator can be decomposed into Kullback–Leibler divergence minus two Jensen-Shannon divergence when discriminator trained to be optimal, which accounts for training instability and mode dropping during GANs training. (Metz et al., 2016) proposed to unroll the optimization of discriminator as surrogate objective to guide update of the generator, which shows improvement of training stability with relatively large computation overhead. (Kodali et al., 2018) claimed that the existence of undesirable local equilibria is responsible for mode collapse and proposed to regularize the discriminator around real data samples with gradient penalty.

Integral probability metric (IPM) based GANs such as Wasserstein GAN (Arjovsky et al., 2017) and its variants (Gulrajani et al., 2017; Wu et al., 2018) can solve gradient vanishing in GANs training theoretically but it is not simple to make discriminator 1-Lipschitz required by the duality conversion practically. Wasserstein GAN (Arjovsky et al., 2017) suggests earth-mover (EM) distance as a measure of discrepancy between two distributions and adopts weight clip to make the discriminator 1-Lipschitz constrained. WGAN-GP (Gulrajani et al., 2017) adopts gradient penalty to regularize discriminator in a less rigorous way, but it requires calculation of the second order derivative with remarkable computation overhead. Spectral normalization on the weight of discriminator proposed by (Miyato et al., 2018) can make discriminator 1-Lipschitz constrained efficiently, but capacity of discriminator is significantly constrained.

Adversarial vulnerability is an intriguing property of neural network-based classifier (Szegedy et al., 2014). A well-trained neural network can give totally wrong prediction to adversarial samples that human can recognize accurately. Small-magnitude adversarial perturbation added to benign data can be easily calculated based on gradient (Goodfellow et al., 2015; Carlini & Wagner, 2017; Dong et al., 2018). (Goodfellow et al., 2015) proposed to augment training data with adversarial samples to improve the robustness of neural networks, which can smooth the decision boundary of classifier around training samples. Gradient of adversarially trained classifier contains more semantic information and less adversarial noise (Tsipras et al., 2018; Kim et al., 2019).

Some work tried to craft or defense against adversarial samples using GANs. (Xiao et al., 2018) proposed to generate adversarial samples efficiently with GANs, in which a generator is used to generate adversarial perturbation for target classifier given original samples. (Shen et al., 2017) proposed AE-GAN to eliminate adversarial perturbation in an adversarial training manner, which can output images with better perceptual quality. Different from their motivations, our work aims at improving the robustness of discriminator by introducing adversarial training on real samples, which does not exist in classic GANs training framework. Similar to our motivation, (Liu & Hsieh) proposed to perform projected gradient attack in supervised GAN training to robustify discriminator. In addition to that, our work clarifies that standard GAN training is approximately equivalent to adversarial training on fake samples. Moreover, we incorporate the proposed adversarial training scheme into unsupervised GAN and achieve improvement of FID score by a large margin.

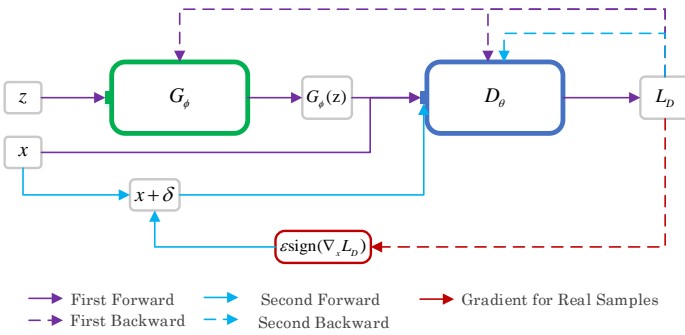

Figure 2: Schematic of proposed AS-GAN. Standard GAN training is illustrated as the first forward pass and the first backward pass. In addition to standard GAN training procedure, we introduce adversarial training of discriminator on real samples, illustrated as the second forward pass and the second backward pass, which is equivalent to train discriminator with robust optimization.

## 3 METHOD

### 3.1 CLASSIC GAN TRAINING FRAMEWORK

In GAN training framework proposed by (Goodfellow et al., 2014), the generator $G_\phi(z)$ parameterized by $\phi$ tries to generate fake samples to fool discriminator and the discriminator $D_\theta(x)$ parameterized by $\theta$ tries to distinguish generated samples between real samples. The formulation using min-max optimization is as follows:

$$\min_\phi \max_\theta V(\theta, \phi) \tag{1}$$

where $V(\theta, \phi)$ is the objective function. Equation 1 can be formulated as a binary classification problem with cross entropy loss:

$$V(\theta, \phi) = \mathbb{E}_{x \sim P_{data}} \left[ \log D_\theta(x) \right] + \mathbb{E}_{z \sim \mathcal{N}(0, I)} \left[ \log D_\theta(1 - G_\phi(z)) \right] \tag{2}$$

where $P_{data}$ is real data distribution and noise $z$ obeys standard Gaussian distribution. When discriminator is trained to be optimal, the training objective for the generator can be reformulated as

Jensen-Shannon divergence, which can measure dissimilarity between two distributions. In practice, we use mini-batch gradient descent to optimize generator and discriminator alternatively.

At each iteration, update rule can be derived as follows:

$$\theta' = \theta + \eta_\theta \nabla_\theta V_m(\theta, \phi, x, \boldsymbol{z}) \tag{3}$$

$$\phi' = \phi - \eta_\phi \nabla_\phi V_m(\theta, \phi, x, \boldsymbol{z}) \tag{4}$$

where $\eta_\theta$ and $\eta_\phi$ are the learning rate of discriminator and generator, respectively. $V_m(\theta, \phi, x, \boldsymbol{z})$ denotes the objective function of mini-batch with $m$ real samples and $m$ fake samples, which is:

$$V_m(\theta, \phi, x, \boldsymbol{z}) = \frac{1}{m} \sum_{i=1}^{m} \left[ \log D_\theta(x^i) + \log D_\theta(1 - G_\phi(\boldsymbol{z}^i)) \right] \tag{5}$$

After updating parameters of networks, fake samples generated by $G_{\phi'}(\boldsymbol{z})$ are adjusted as following equation according to chain rule:

$$G_{\phi'}(\boldsymbol{z}) \approx G_\phi(\boldsymbol{z}) - \eta_\phi \frac{\partial G_\phi(\boldsymbol{z})}{\partial \phi} \nabla_\phi V_m(\theta, \phi, x, \boldsymbol{z}) \tag{6}$$

$$= G_\phi(\boldsymbol{z}) - \eta_\phi \frac{\partial G_\phi(\boldsymbol{z})}{\partial \phi} \left( \frac{\partial G_\phi(\boldsymbol{z})}{\partial \phi} \right)^T \nabla_{G_\phi(\boldsymbol{z})} V_m(\theta, \phi, x, \boldsymbol{z}) \tag{7}$$

where $\frac{\partial G_\phi(\boldsymbol{z})}{\partial \phi}$ is a Jacobian matrix. The updated $G_{\phi'}(\boldsymbol{z})$ can be seen as adversarial samples of the discriminator at this iteration because $\eta_\phi$ is usually small. These samples will be fed into the discriminator at future iteration to perform adversarial training. From this point of view, the classic training framework mainly includes adversarial training on fake samples, which is illustrated as the first pass in Figure 2. Nevertheless, adversarial training of discriminator on real samples does not exist in this framework, which makes adversarial training unsymmetric and unbalanced. Adversarial noise contained in the gradient of non-robust discriminator can make training unstable because of the unsmoothed decision boundary of discriminator around real data.

## 3.2 ROUBUST OPTIMIZATION

In order to make discriminator more robust, we propose the following objective for robust optimization of discriminator:

$$V(\theta, \phi) = \mathbb{E}_{x \sim P_{data}} \left[ \min_{\|\delta\|_p \leq \varepsilon} \log D_\theta(x - \delta) \right] + \mathbb{E}_{\boldsymbol{z} \sim \mathcal{N}(0, \boldsymbol{I})} \left[ \log D_\theta(1 - G_\phi(\boldsymbol{z})) \right] \tag{8}$$

In fact, when training data of real samples is infinite, the above objective is approximately equivalent to the original. However, in practice, training data is always limited, which partly accounts for the existence of adversarial samples. With the proposed objective, discriminator is not only required to classify real data correctly but also should not be vulnerable to small perturbation. This robust optimization training scheme can smooth the decision boundary of discriminator and prevent it from being stuck in a local minimum.

## 3.3 ADVERSARIAL TRAINING ON REAL SAMPLES

In this paper, we select adversarial training as implementation to solve the proposed robust optimization problem with $L_\infty$-norm constraint. Specifically, we introduce adversarial training on real data that does not exist in the original framework, which can make adversarial training symmetric and stabilize GAN training. Practically, we perform adversarial training after Equation 3 at each iteration as the following equation:

$$\theta' = \theta + \eta_\theta \nabla_\theta V_m(\theta, \phi, \hat{x}, \boldsymbol{z}) \tag{9}$$

where $\hat{x}$ is an adversarial sample of discriminator, perturbation of which can be obtained by backward propagation of Equation 5 with respect to $x$ with negligible computation overhead. The adversarial sample can be calculated with constant $\varepsilon$ as follows:

$$\hat{x} = x - \varepsilon \operatorname{sign}(\nabla_x V_m(\theta, \phi, x, \boldsymbol{z})) \tag{10}$$

This adversarial training formulation is adopted from (Goodfellow et al., 2015), which calculates $L_\infty$-norm constrained perturbation by linearizing objective function. Adversarial training on real samples of discriminator is illustrated as the second pass in Figure 2, where $L_D$ denotes the minus of $V_m(\theta, \phi, x, z)$. We make adversarial training procedure symmetric by performing adversarial training both on fake samples and real samples. In this way, discriminator can provide more informative gradient with less adversarial noise, which can stabilize training and accelerate convergence. Please refer to Algorithm 1 for more details about symmetric adversarial training.

### 3.4 Effective magnitude of perturbation

It is crucial to set an appropriate magnitude of perturbation to make adversarial training effective. When perturbation set to zero, the proposed method degrades to updating discriminator twice on the same real data. When perturbation set to a too large value, real data will be drastically perturbed. Semantic information and quality will be changed, which can mislead discriminator to recognize degraded samples as real data incorrectly. In a sense, adversarial training on real samples regularizes discriminator by augmenting training data, which prevents discriminator from being too strong to alleviate training collapse. The capability of two networks become more balanced with adversarial training both on real samples and fake samples. Noteworthily, the network capacity of discriminator should be larger or at least equivalent to that of generator because discriminator is further constrained with additional adversarial training. In addition, we suggest to set pertubation to zero in the begining of training in case discriminator is too weak. We do extensive experiments on how perturbation affects training in the next section.

## 4 Experiments

For the purpose of evaluating our method and investigating the reason behind its efficacy, we test our adversarial training method on image generation tasks on CIFAR-10, CelebA and LSUN with DCGAN and ResNet architecture. CIFAR-10 is a well-studied dataset of 32 x 32 natural images, containing 10 categories of objects and CelebA is a large-scale face attributes dataset with more than 200k images. In addition, We validate the proposed method on 3000k images labeled as bedroom in LSUN . For fast validation, We resize images in CelebA and LSUN to 64 x 64. The effectiveness of the proposed method can be proved empirically by results on these distinct datasets with two widely adopted architectures.

In section 4.1, we first conduct hyper-parameter experiments on the magnitude of perturbation for unsupervised generation task on CIFAR-10. In order to study transferability on different network structure, we conduct some extensive experiments with multiple architectures and settings in section 4.2, which also demonstrate the advantages of our method that can stabilize training and accelerate convergence in section 4.3. Lastly, we compare our adversarial symmetric training method with some latest state-of-the-art GAN variants. With the same experimental settings of some classic models, our method can produce diverse images with better fidelity.

In this paper, we use standard GAN objective function as the adversarial loss. In order to alleviate gradient vanishing at the beginning of training, we adopt non-saturating loss suggested by (Goodfellow et al., 2014). Furthermore, we use *Fréchet inception distance* (FID)(Heusel et al., 2017) and *inception score*(Salimans et al., 2016) to measure model performance, both of which are well-studied metric of image quality. Please refer to Appendix F for more details about implementation. Our source code will be released on Github.

### 4.1 Evaluation with different hyper-parameters

In order to find an appropriate magnitude of adversarial perturbation, we do unsupervised image generation experiments on CIFAR-10 with DCGAN architecture at different settings of perturbation. Due to the large searching space, we select several typical values for experiments such as $\{0,1,2,3,4\}/255$. Furthermore, we do ablation study by replacing gradient used to craft perturbation by Gaussian noise. All experiments are run three times independently to reduce randomness.

Figure 3 shows the FID results in the different hyper-parameter setting, Our method performs far better than baseline when $\varepsilon$ lies in interval $0.5/255 \sim 3/255$. However, no matter the perturbation

amplitude is too small ($\varepsilon = 0.2/255$) or a little large ($\varepsilon = 4/255$), the method improves original model slightly. In addition, when the imposed perturbation is too strong, the model performs even worse than the baseline. This is because samples perturbed drastically can mislead discriminator to recognize degraded samples as real data incorrectly, which hinders discriminator from catching true characteristic of real data distribution. With appropriate perturbation, discriminator can be regularized to balance training, which can alleviate the bothersome collapse of training loss and generate samples in standard GAN. Only when the attacking intensity is suitable, discriminator will become more robust owning to adversarial training, facilitating itself to produce more accurate and informative gradient to generator, which can promote the capability of generator tremendously. In this way, generator obtains more meaningful and reliable update information, accounting for generating better samples. Still and all, too much perturbation is harmful to discriminator when exceeding its endurance, leading discriminator to the ground defeated by generator, which is not a favorable scene. Similarly, when the imposed perturbation is too tiny, on which condition the adversarial sample of real data is almost close to the benign, the effect of adversarial training is limited. In particular, when $\varepsilon$ is zero, the model degrades into regular GAN algorithm, the only change of which is updating discriminator twice on same real data. This setting is only slightly better than the baseline but far worse than the version with appropriate $\varepsilon$. Overall, results show that discriminator adversarially trained with appropriate perturbation can stabilize training and improve performance of generator.

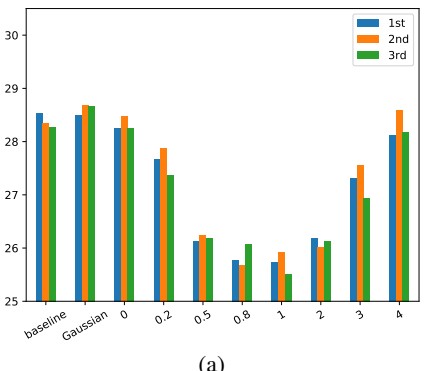 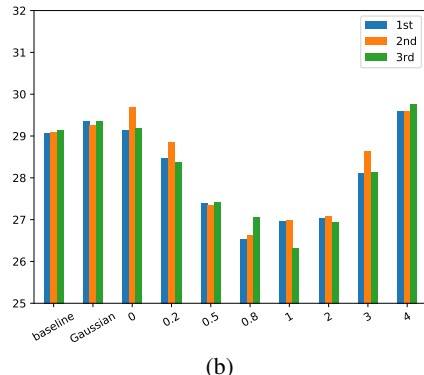

(a)                                        (b)

Figure 3: (a) Best FID results under different settings of three independent runs (Lower is better). (b) Mean FID of last 20 epoch. With appropriate perturbation and adversarial updating policy, the quality of generated samples can go beyond baseline by a large margin.

Besides, the annotation 'Gaussian' in Figure 3 means that the gradient used to craft adversarial samples is replaced by Gaussian noise, whose poor performance proves that perturbation direction of gradient is important for effectiveness. In the following sections, we use $\varepsilon = 1/255$ as default option for simplicity.

### 4.2 EVALUATION WITH DIFFERENT ARCHITECTURES

To explore the transferability and compatibility of the proposed method, we test with widely adopted DCGAN and ResNet architecture on CIFAR-10 and CelebA. Figure 4 plots the comparison results. Additionally, two groups of contrast experiments with different setttings are performed (see Appendix E). We find that our method can largely accelerate convergence and improve the fidelity of generated samples. By applying our method, stability can also be improved vastly (Appendix A.1). Even with the setting in which baseline GAN collapse thoroughly, our model can still converge toward optimization objective.

Generally, GANs collapse when discriminator is too strong or too weak; the former situation is more common during training. WGAN and a series of its variants solve the problem theoretically by using Wasserstein distance to measure the discrepancy between two distributions with constraining the

discriminator. Different from their approach, we address training instability in a practical point of view that discriminator as a classifier is not ideal and robust. Adversarial training on real samples is introduced into classific GAN to make discriminator more robust. Experimental results prove our method is effective with marginal computation cost and can be applied to different network architectures on different datasets.

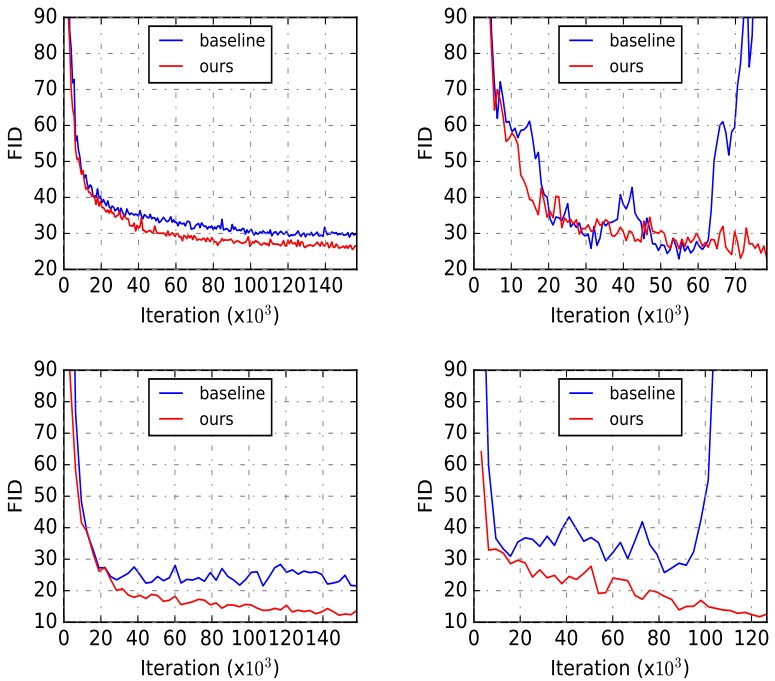

Figure 4: Training curves of FID on CIAFR-10 (upper) and CelebA (lower) with DCGAN (left) and ResNet (right). Results show that the proposed method can accelerate convergence and achieve better FID. Meanwhile, it can stabilize training with less sensitivity to network architecture and hyper-parameter setting.

### 4.3 IMPROVED PERFORMANCE

We perform image generation experiments to verify the effectiveness of our method. As shown in Table 2, through unsupervised training on CIFAR-10, CelebA and bedroom in LSUN, our model can achieve comparable performance to state of the art on FID and inception score (Higher is better). It should be noted that the four rows at the bottom show the results of our implementation. We further validate on ImageNet (Russakovsky et al., 2015) resized to 64 x 64 with unsupervised AS-DCGAN and achieve a FID of 60.65 relative to 65.78 of baseline. Generated samples are shown in Appendix C.1.

### 4.4 ANALYSIS

To show how our method works, we perform some analytical experiments on CIFAR-10. Notice that in our algorithm, the only significant change for vanilla GAN is adversarial training on real samples during the two-players game. Gradient given by discriminator is the key to update generator. Hence, we visualize the gradient as Figure 1, which shows that the gradient of adversarially trained discriminator contains more semantic information, eg, profiles of face, but the gradient of standard discriminator looks like uninformative noise. We further show the histogram of the gradient of discriminator with respect to real samples as Figure 5a. Our method can obtain more sparse gradient

Table 1: Inception scores and FIDs with unsupervised image generation on CIFAR-10 and CelebA. ⋆ (Radford et al., 2015)(experimented by (Yang et al., 2017)), †(Miyato et al., 2018), ‡ (Wu et al., 2017), ∗(Gulrajani et al., 2017)

| Method | Inception score CIFAR-10 | FID CIFAR-10 | CelebA | LSUN |
|---|---|---|---|---|
| (Standard CNN) | | | | |
| DCGAN | $6.64\pm.14^\star$ | $30.9^\ddagger$ | $52.0^\ddagger$ | $61.1^\ddagger$ |
| WGAN-GP | $6.68\pm.06^\dagger$ | $40.2^\dagger$ | $21.2^\ddagger$ | |
| SN-GAN$^\dagger$ | $7.58\pm.12$ | $25.5$ | | |
| WGAN-GP(ResNet) | $7.86\pm.07^*$ | $18.8^\ddagger$ | $18.4^\ddagger$ | $26.8^\ddagger$ |
| WGAN-div(ResNet)$^\ddagger$ | | $18.1$ | $15.2$ | $15.9$ |
| DCGAN | $7.05\pm.14$ | $28.05$ | $20.45$ | $25.36$ |
| (ours)AS-DCGAN | $\mathbf{7.21}\pm.02$ | $\mathbf{25.50}$ | $\mathbf{10.90}$ | $\mathbf{18.08}$ |
| ResNet | $7.35\pm.16$ | $22.92$ | $25.72$ | $175.70$ |
| (ours)AS-GAN(ResNet) | $\mathbf{7.65}\pm.15$ | $\mathbf{21.84}$ | $\mathbf{11.71}$ | $\mathbf{45.96}$ |

and lower L1 norm (Figure 5b) as training iteration increases, which means adversarial noise in gradient is partly eliminated. What's more, the proposed method can augment training data and smooth the decision boundary of discriminator, which can alleviate mode collapse to a large extent shown as Appendix C.2. By means of adversarial training evenly on both real and fake samples, we make the training scheme symmetric and stable.

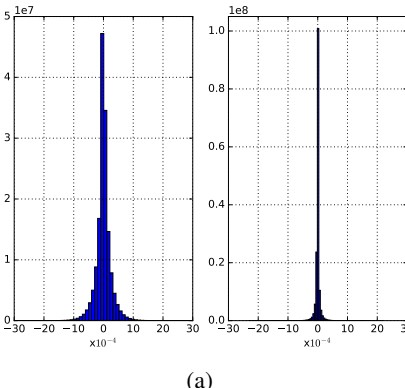 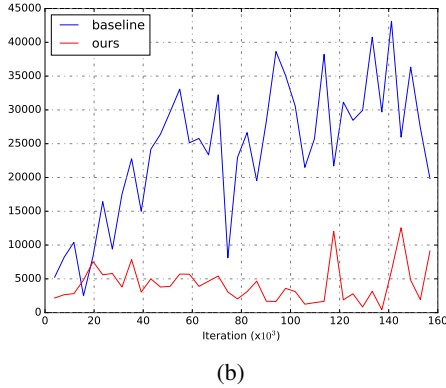

(a)  (b)

Figure 5: (a) Histogram of gradient from discriminator with respect to input images. The left is the baseline, and the right is the adversarial symmetrically trained discriminator. (b) L1-norm evolution of discriminator gradient during training. Adversarial training will prompt the gradient becoming more sparse.

The computation overhead of the proposed method is about 25% relative to the baseline, which is comparable to that of spectral normalization and much smaller than that of gradient penalty. Comparison of average training time of one epoch of different methods is shown as Table 4.4.

Table 2: Average training time of different methods

| Setting | DCGAN | (ours)AS-DCGAN | SN-DCGAN | DCGAN-GP |
|---|---|---|---|---|
| Training time | 19.83s | 26.40s | 24.50s | 31.57s |

## 5 CONCLUSION

The relationship between GANs and adversarial samples has been a open question since both models emerged. In this paper, we analyze that adversarial training on fake samples is already taken into account in standard GAN training framework, but adversarial training on real samples does not exist, which can make training unbalanced and unstable. This is because gradient given by non-robust discriminator contains more adversarial noise, which can mislead update of the generator. In order to make training scheme symmetric and discriminator robust, we introduce adversarial training on real samples. We validate the proposed method on image generation tasks on CIFAR-10, CelebA and bedroom in LSUN with varied network architectures. Experiments show that the gradient of discriminator adversarially trained both on real and fake samples contains less adversarial noise. Convergence speed and performance are improved with marginal computation overhead. Moreover, mode collapse is alleviated. With simple DCGAN network architecture and standard objective function, we can achieve comparable FID to state of the art on these datasets.

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

## A  ANALYSIS

### A.1  TRAINING STABILITY

We further demonstrate the superiority of our algorithm over traditional GAN methods about training stability. During training on CIFAR-10, we plot the prediction confidence of discriminator about real data and adversarial samples of real data. The detailed confidence and loss value of discriminator with the training process are depicted as Figure 6.

It clearly exhibits that the discriminator of traditional DCGAN has very unstable output confidence and loss curve during training. With our method, both of the variables $D_\theta(x)$ and $errD$ become more smooth and stable. Moreover, when adversarial perturbation $\varepsilon$ is appropriate for training, the confidence about adversarial samples $D_\theta(\hat{x})$ will be basically lower than $D_\theta(x)$ by a large margin. Because in the beginning, the discriminator is non-robust and sensitive to adversarial samples. With more iterations, discriminator will become robust to adversarial samples, thus $D_\theta(\hat{x})$ converges to be higher than before, but still inferior to $D_\theta(x)$ due to non-eliminated perturbation. As shown in Appendix D, discriminator with adversarial training is more robust than standard discriminator. Similarly, the loss is also stabilized with our algorithm (Figure 6b), leading to a more easy-training, less-sensitive to hyper-parameters and high-performance model.

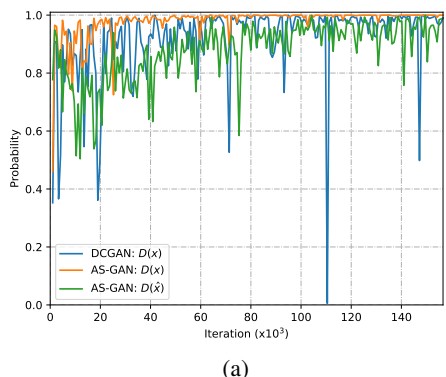 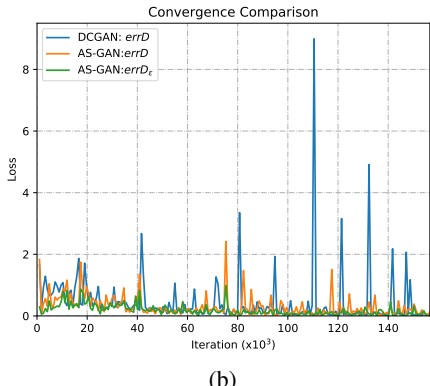

         (a)                                       (b)

Figure 6: (a) Confidence of discriminator on real data and adversarial samples of real data during training. (b) Evolution of discriminator loss at different settings.

## A.2 ABLATION STUDY

In addition, we visualize the convergence process through evaluation by inception score and FID. As described in Figure 7, when updating discriminator on real samples twice ($\varepsilon = 0$) or giving the perturbation of a random direction ('Gaussian'), the models achieve similar performance to baseline. Only with the appropriate setting of perturbation and symmetric training scheme, desirable results can be realized.

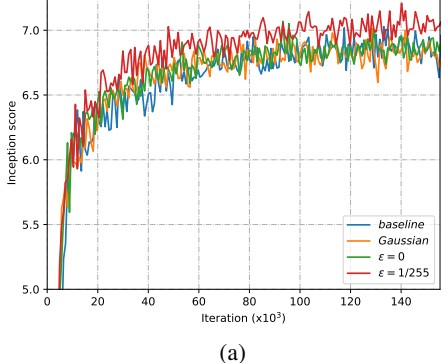 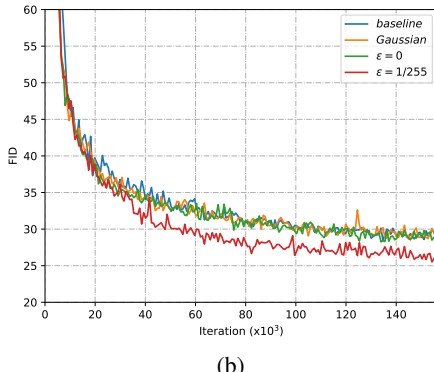

         (a)                                       (b)

Figure 7: (a) Training curves of inception score with different methods. (b) Training curves of FID with different methods. With the increasing of iterations, our algorithm ($\varepsilon = 1/255$) converges faster and better than other models.

## B  ALGORITHM OF AS-GAN

**Algorithm 1** Mini-batch stochastic gradient descent training of AS-GAN. We set perturbation $\varepsilon$ to $1/255$ as default for image generation tasks.

1: **for** number of training iterations **do**
2:     Sample mini-batch of $m$ noise samples $\{\boldsymbol{z}^i, ..., \boldsymbol{z}^m\}$ from Gaussian distribution $\mathcal{N}(0, \boldsymbol{I})$.
3:     Sample mini-batch of $m$ data samples $\{x^i, ..., x^m\}$ from real data distribution $P_{data}$.
4:     Update the discriminator by gradient ascent, this step can be regarded as adversarial training of discriminator on fake samples:

$$\theta' = \theta + \eta_\theta \nabla_\theta V_m(\theta, \phi, x, \boldsymbol{z}) \tag{11}$$

5:     **Craft adversarial samples of real samples for discriminator based on gradient**:

$$\hat{x} = x - \varepsilon \operatorname{sign}(\nabla_x V_m(\theta, \phi, x, \boldsymbol{z})) \tag{12}$$

6:     **Perform adversarial training of discriminator on real samples**:

$$\theta' = \theta + \eta_\theta \nabla_\theta V_m(\theta, \phi, \hat{x}, \boldsymbol{z}) \tag{13}$$

7:     Update generator by gradient descent, this step can be regarded as crafting adversarial samples of fake data:

$$\phi' = \phi - \eta_\phi \nabla_\phi V_m(\theta, \phi, x, \boldsymbol{z}) \tag{14}$$

8: **end for**

## C  RESULTS PRESENTATION

### C.1  GENERATED SAMPLES

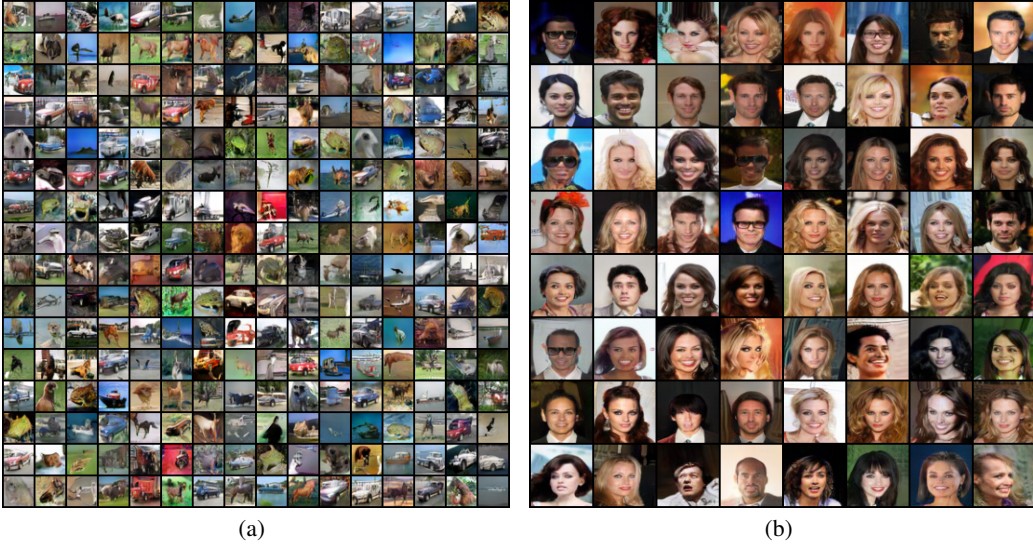

(a)                                        (b)

Figure 8: (a): 32 x 32 CIFAR-10 samples generated by AS-ResNet. (b): 64 x 64 CelebA samples generated by AS-ResNet. We believe these samples are at least comparable to the best published results so far.

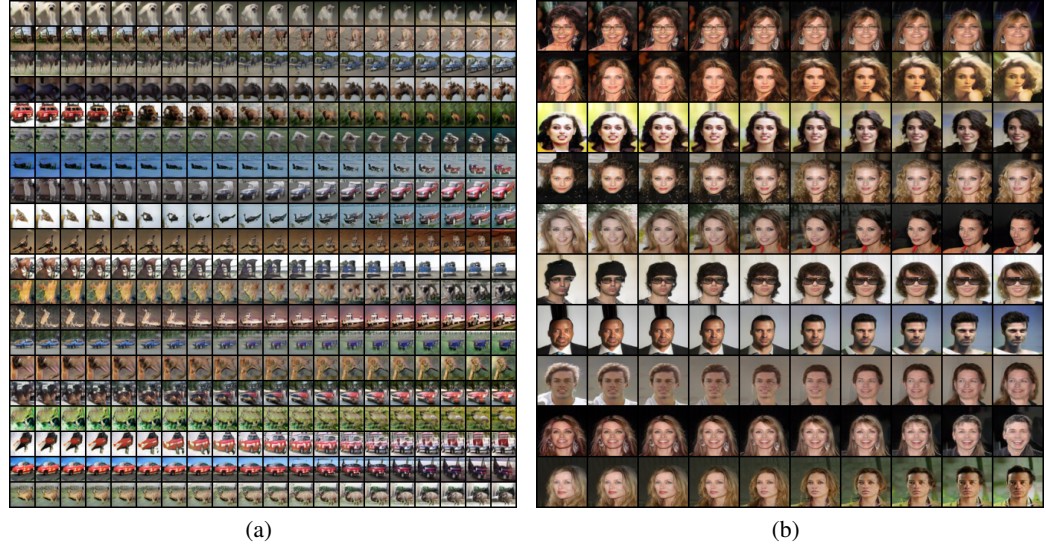

(a)                                    (b)

Figure 9: (a): Interpolation results from AS-ResNet on CIFAR-10. (b): Interpolation results from AS-ResNet on CelebA.

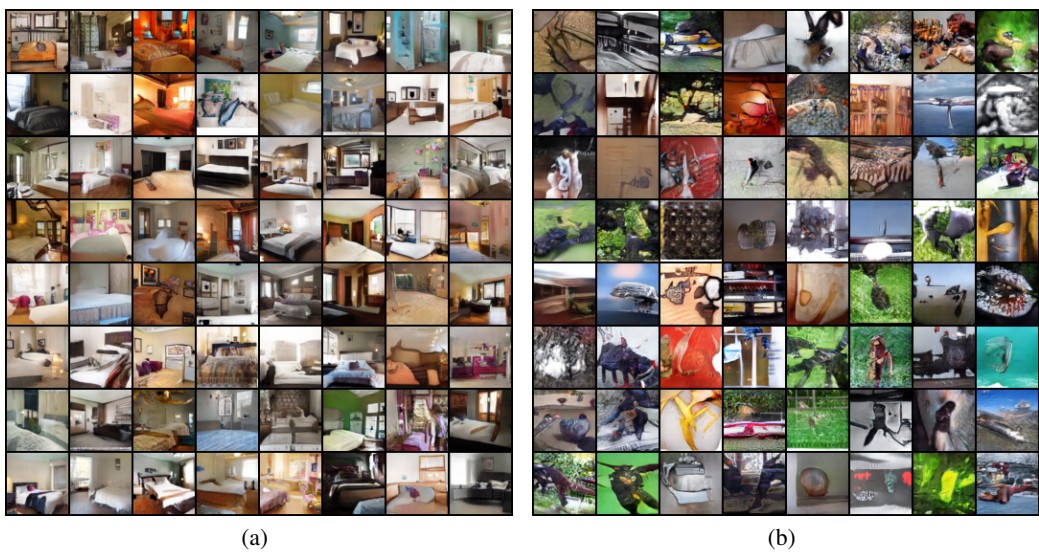

(a)                                    (b)

Figure 10: (a): 64 x 64 results from AS-DCGAN on bedroom in LSUN. (b): 64 x 64 results from unsupervised AS-DCGAN on ImageNet.

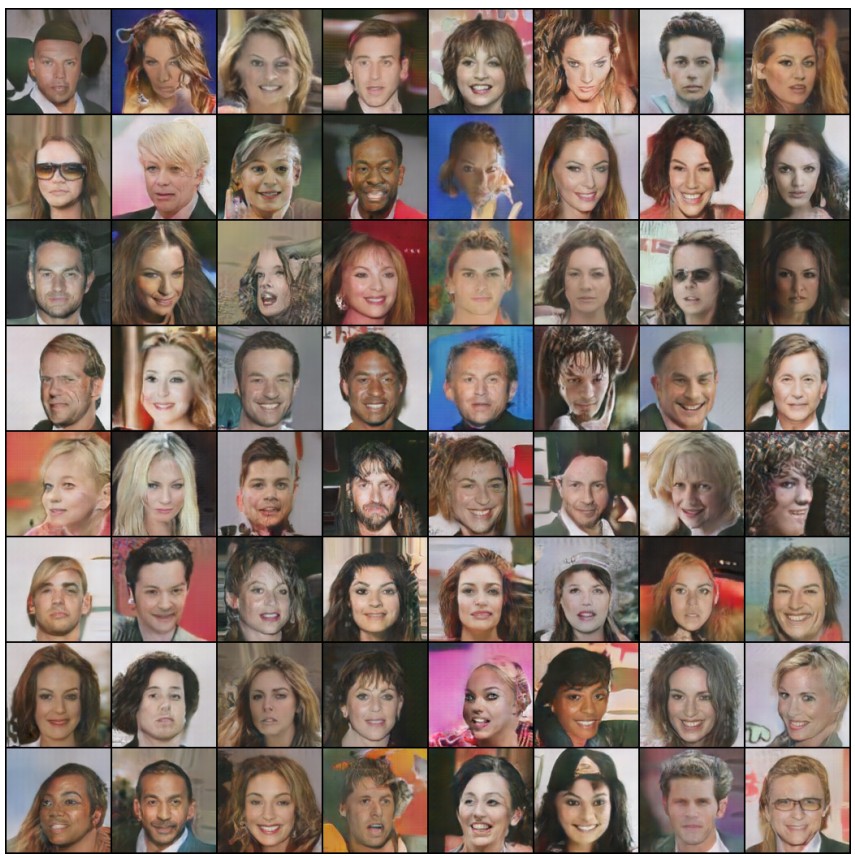

Figure 11: 128 x 128 results from AS-DCGAN on CelebA.

## C.2 ALLEVIATED MODE COLLAPSE

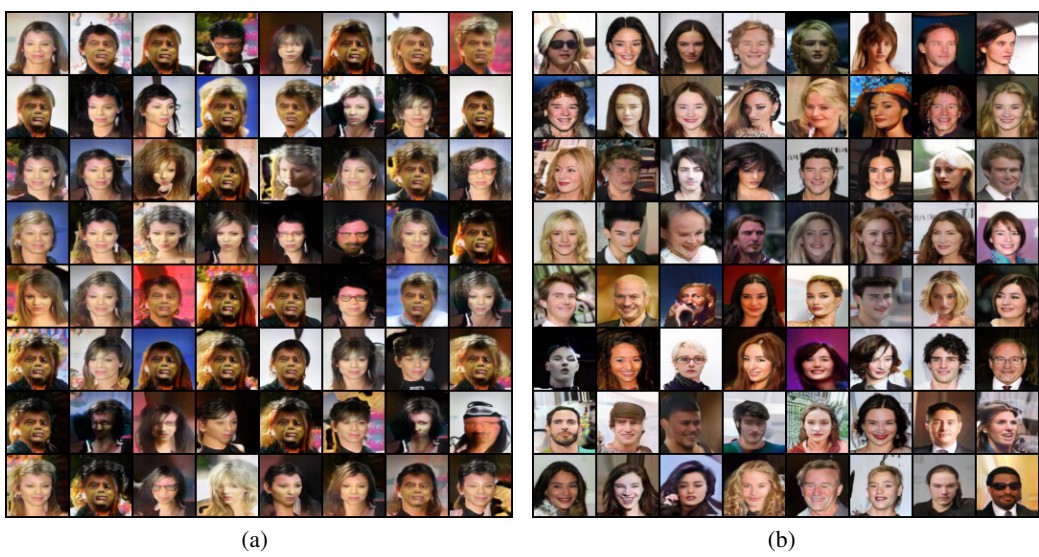

(a)                                      (b)

Figure 12: (a): Collapsed samples generated by standard GAN trained on CelebA. (b): Samples generated by AS-ResNet trained on CelebA.

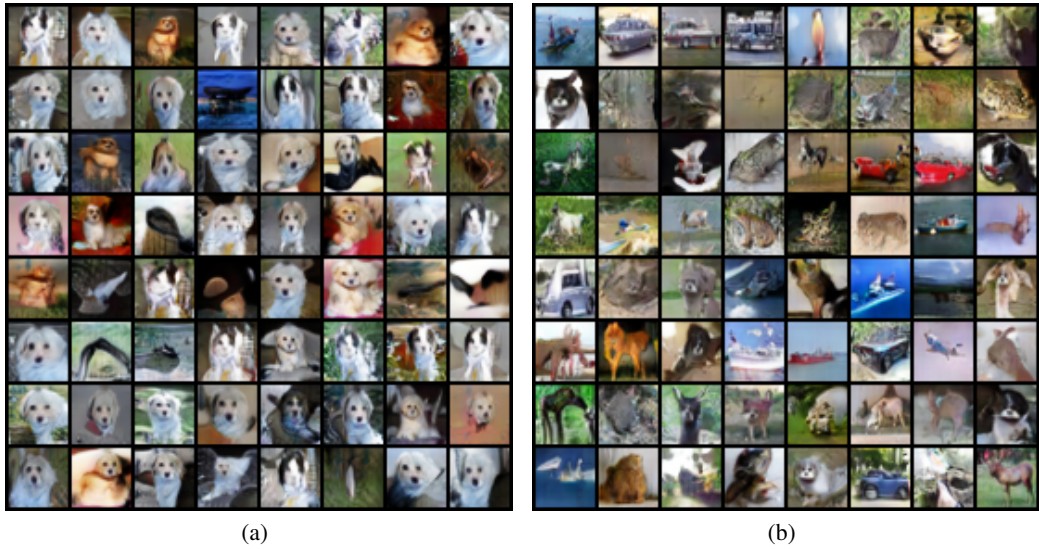

(a) (b)

Figure 13: (a): Collapsed samples generated by standard GAN trained on CIFAR-10. (b): Samples generated by AS-ResNet trained on CIFAR-10.

# D  ADVERSARIAL SAMPLES OF DISCRIMINATOR

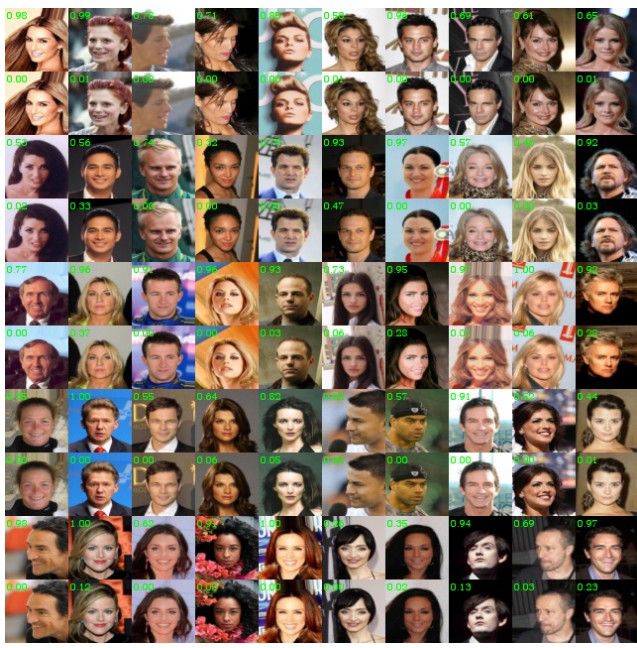

Figure 14: Benign samples (on odd rows) and adversarial samples of standard discriminator (on even rows). Confidence is depicted at corner. Standard discriminator is extremely vulnerable to imperceptible perturbation. The pertubation level is 1/255.

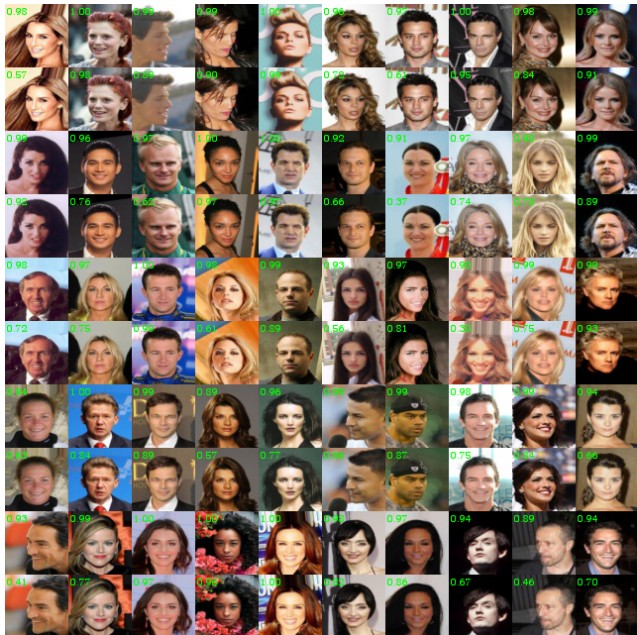

Figure 15: Benign samples (on odd rows) and adversarial samples of discriminator adversarially trained on real data (on even rows). Confidence is depicted at corner. After adversarial training on real samples, discriminator is more robust to adversarial perturbation. The pertubation level is 1/255.

# E   COMPARISON IN DIFFERENT SETTINGS

Table 3: FIDs with unsupervised image generation on CIFAR-10.

| Setting | Standard | Adam($\beta_2$=0.9) | no BN(G) | no BN(D) | no BN(Both) | $lr_D$=4e−4 |
|---|---|---|---|---|---|---|
| Baseline | 28.05 | 29.14 | 29.40 | 69.82 | 59.15 | 28.44 |
| AS-GAN(ours) | 25.50 | 27.18 | 26.72 | 51.30 | 56.18 | 26.57 |

Here, 'standard' means using same setting with Appendix F. From table 3 and table 4, under different conditions, our model is always better than baseline.

Table 4: FIDs with unsupervised image generation on CelebA.

| Setting | $lr_D$=2e−4 $lr_G$=2e−4 | $lr_D$=5e−5 $lr_G$=2e−4 |
|---|---|---|
| Baseline | 34.10 | 18.73 |
| AS-GAN(ours) | 12.14 | 11.82 |

# F   IMPLEMENTATION AND TRAINING DETAILS

Our models are implemented by Pytorch with acceleration of RTX 2080ti GPUs, only when evaluating inception score we use Tensorflow. Before fed into discriminator, we rescale images within $[-1, 1]$. Dimension of the latent vector is set to 100 for all implementations. We train models on CelebA for 100 epoch, CIFAR-10 for 200 epoch, LSUN for 10 epoch and Imagenet for 10 epoch. FID and inception score are computed from 22400 generated samples, and inception score is calculated by 10 independent partitions.

In ResNet architecture, the residual block is organized as BatchNorm-ReLU-Resample-Conv-BatchNorm-ReLU-Conv with skip connection. We use bilinear interpolation for upsampling and average pooling for downsampling. Batch normalization (Ioffe & Szegedy, 2015) is used both for generator and discriminator. Parameters of network are initialized by Xavier method. We train networks with Adam optimizer with learning rate 2e-4. $\beta_1$ is set to 0.5 and $\beta_2$ is set to 0.999.

In DCGAN architecture, the basic block is organized as Conv-BatchNorm-LeakyReLU for discriminator or ConvTransposed-BatchNorm-ReLU for generator. The weights of convolution are initialized by normal disitribution with zero mean and 0.02 standard deviation. We do not use bias in convolution. We train DCGAN with Adam optimizer with learning rate 2e-4. $\beta_1$ is set to 0.5 and $\beta_2$ is set to 0.999. Because training standard GAN on CelebA is unstable, we decrease learning rate of discriminator to 5e-5 to balance training as TTUR training strategy (Heusel et al., 2017).

Table 5: ResNet Generator $G_\phi(\boldsymbol{z})$ for 32 x 32

| Block | Kernel size | Resample | Output shape |
|---|---|---|---|
| Input $\boldsymbol{z}$ | | | 100 |
| Linear | | | $256 \times 4 \times 4$ |
| Residual block | $[3 \times 3] \times 2$ | Up | $256 \times 8 \times 8$ |
| Residual block | $[3 \times 3] \times 2$ | Up | $256 \times 16 \times 16$ |
| Residual block | $[3 \times 3] \times 2$ | Up | $256 \times 32 \times 32$ |
| Conv, tanh | $3 \times 3$ | | $3 \times 32 \times 32$ |

Table 6: ResNet Discriminator $D_\theta(x)$ for 32 x 32

| Block | Kernel size | Resample | Output shape |
|---|---|---|---|
| Residual block | $[3 \times 3] \times 2$ | Down | $256 \times 16 \times 16$ |
| Residual block | $[3 \times 3] \times 2$ | Down | $256 \times 8 \times 8$ |
| Residual block | $[3 \times 3] \times 2$ | Down | $256 \times 4 \times 4$ |
| Residual block | $[3 \times 3] \times 2$ | Down | $256 \times 2 \times 2$ |
| ReLU, Average pool | | | 256 |
| Linear, Sigmoid | | | 1 |

Table 7: ResNet Generator $G_\phi(\boldsymbol{z})$ for 64 x 64

| Block | Kernel size | Resample | Output shape |
|---|---|---|---|
| Input $\boldsymbol{z}$ | | | 100 |
| Linear | | | $256 \times 4 \times 4$ |
| Residual block | $[3 \times 3] \times 2$ | Up | $256 \times 8 \times 8$ |
| Residual block | $[3 \times 3] \times 2$ | Up | $256 \times 16 \times 16$ |
| Residual block | $[3 \times 3] \times 2$ | Up | $256 \times 32 \times 32$ |
| Residual block | $[3 \times 3] \times 2$ | Up | $256 \times 64 \times 64$ |
| Conv, tanh | $3 \times 3$ | | $3 \times 64 \times 64$ |

Table 8: ResNet Discriminator $D_\theta(x)$ for 64 x 64

| Block | Kernel size | Resample | Output shape |
|---|---|---|---|
| Residual block | $[3 \times 3] \times 2$ | Down | $256 \times 32 \times 32$ |
| Residual block | $[3 \times 3] \times 2$ | Down | $256 \times 16 \times 16$ |
| Residual block | $[3 \times 3] \times 2$ | Down | $256 \times 8 \times 8$ |
| Residual block | $[3 \times 3] \times 2$ | Down | $256 \times 4 \times 4$ |
| Residual block | $[3 \times 3] \times 2$ | Down | $256 \times 2 \times 2$ |
| ReLU, Average pool | | | 256 |
| Linear, Sigmoid | | | 1 |

Table 9: DCGAN Generator $G_\phi(\boldsymbol{z})$ for 32 x 32

| Block | Kernel size | Stride | Output shape |
|---|---|---|---|
| Input $\boldsymbol{z}$ | | | 100 |
| Basic block | $4 \times 4$ | 1 | $512 \times 4 \times 4$ |
| Basic block | $4 \times 4$ | 2 | $256 \times 8 \times 8$ |
| Basic block | $4 \times 4$ | 2 | $128 \times 16 \times 16$ |
| ConvTransposed, tanh | $4 \times 4$ | 2 | $3 \times 32 \times 32$ |

Table 10: DCGAN Discriminator $D_\theta(x)$ for 32 x 32

| Block | Kernel size | Stride | Output shape |
|---|---|---|---|
| Conv, LeakyReLU | $4 \times 4$ | 2 | $128 \times 16 \times 16$ |
| Basic block | $4 \times 4$ | 2 | $256 \times 8 \times 8$ |
| Basic block | $4 \times 4$ | 2 | $512 \times 4 \times 4$ |
| Conv, Sigmoid | $4 \times 4$ | 1 | 1 |

Table 11: DCGAN Generator $G_\phi(\boldsymbol{z})$ for 64 x 64

| Block | Kernel size | Stride | Output shape |
|---|---|---|---|
| Input $\boldsymbol{z}$ | | | 100 |
| Basic block | $4 \times 4$ | 1 | $1024 \times 4 \times 4$ |
| Basic block | $4 \times 4$ | 2 | $512 \times 8 \times 8$ |
| Basic block | $4 \times 4$ | 2 | $256 \times 16 \times 16$ |
| Basic block | $4 \times 4$ | 2 | $128 \times 32 \times 32$ |
| ConvTransposed, tanh | $4 \times 4$ | 2 | $3 \times 64 \times 64$ |

Table 12: DCGAN Discriminator $D_\theta(x)$ for 64 x 64

| Block | Kernel size | Stride | Output shape |
|---|---|---|---|
| Conv, LeakyReLU | $4 \times 4$ | 2 | $128 \times 32 \times 32$ |
| Basic block | $4 \times 4$ | 2 | $256 \times 16 \times 16$ |
| Basic block | $4 \times 4$ | 2 | $512 \times 8 \times 8$ |
| Basic block | $4 \times 4$ | 2 | $1024 \times 4 \times 4$ |
| Conv, Sigmoid | $4 \times 4$ | 1 | 1 |

