# OpenReview forum: "BRIDGING ADVERSARIAL SAMPLES AND ADVERSARIAL NETWORKS"
_ICLR.cc/2020/Conference — Reject_

### Official Review · AnonReviewer3 · 2019-10-11
**Official Blind Review #3**

**Rating:** 3

**Review:**

The paper proposes a new GAN training that additionally feeds adversarial examples as the real samples to the discriminator D. A key motivation here is to regularize the target real distribution simulated by D to be robust to adversarial perturbations. Experimental results show that the proposed GAN training generally improves the generation performance from the vanilla GAN training in CIFAR-10, CelebA and LSUN datasets.

In overall, I liked its clear motivation and the simplicity of the method. Experimental results are also presented clearly, and shows a significant improvement. One of my main concerns, however, is that robustifying D in GAN training is not a new idea for some readers [1], so they need more clarification on the novelty of the proposed method, e.g. by discussing about it in related work or by comparing the performance.

- In regarding robust optimization, I think [2] could cover a lot of practices considered in this paper. Some questions listed here are relevant to this point:
    (a) The paper only considers to use FGSM in adversarial training part, but FGSM training on large epsilon usually leads to overfitting [2] on CIFAR-10. I wonder if the authors have tried PGD counterpart in their method.
    (b) It seems that the method uses both natural and adversarial examples in adversarial training, as in [3]. Instead, there is another (and perhaps more common) type of adversarial training [2] that uses only adversarial examples for the training. What happens if this training is applied to the method?

- The method makes an additional parameter updates (Ep. 3 and 8) for adversarial training. Could this step make additional gain to the method by training more, i.e. perhaps it is a bit unfair to the vanilla training?

- It would strengthen the claim if the paper could present whether the robustness of D is indeed increased, e.g. by comparing adversarial accuracies?

- Table 1: I feel there should be more discussion about why the reproduced values are so different compared to that of previously reported values, as they might confuse the readers to convince the claimed results.

- Eq. 7: The first term in the right hand side have to be E[max (log D(x - d))] instead of E[(log D(x - d))]?

[1] Liu, X., & Hsieh, C. J. (2019). Rob-gan: Generator, discriminator, and adversarial attacker. CVPR 2019.
[2] Madry, A. et al. (2017). Towards deep learning models resistant to adversarial attacks. ICLR 2018.
[3] Goodfellow, I. et al. (2014). Explaining and harnessing adversarial examples. arXiv preprint arXiv:1412.6572.

**Experience Assessment:**

I have read many papers in this area.

**Review Assessment: Checking Correctness Of Derivations And Theory:**

N/A

**Review Assessment: Checking Correctness Of Experiments:**

I assessed the sensibility of the experiments.

**Review Assessment: Thoroughness In Paper Reading:**

I read the paper at least twice and used my best judgement in assessing the paper.

---

> ### Author Response · Authors · 2019-11-08
> **Response for reviewer#3 part 1**
>
> Thank you for your detailed and thoughtful review.  We will address your concerns one by one below.
>
> 1.’ In overall, I liked its clear motivation and the simplicity of the method. Experimental results are also presented clearly, and shows a significant improvement. One of my main concerns, however, is that robustifying D in GAN training is not a new idea for some readers [1], so they need more clarification on the novelty of the proposed method, e.g. by discussing about it in related work or by comparing the performance.’
>
> Response:
> We will discuss the mentioned work in related work. The motivation of Rob-GAN is similar to ours. The difference are as follows: First, one of our key contribution is that, we proved that standard GAN training is approximately equivalent to adversarial training on fake samples but adversarial training on real samples is missed, so standard discriminator is not robust to adversarial perturbation around real samples. Hence, we introduce adversarial training on real samples to make this procedure more symmetric and make discriminator more robust. However, this is not investigated in Rob-GAN. Second, we focus on unsupervised training and achieve FID improvement by 10%~50% on CIFAR10, CelebA ,and LSUN, But the work [1] focus on supervised training. Moreover, we explore the effect of perturbation level on performance improvement with different network structures and find proper default setting. Third, we adopt FGSM as adversarial training scheme but the work used PGD. In our experiments, we find that adversarial training with FGSM can achieve comparable performance to that with PGD but the computation overhead is much small, because FGSM only need update once. Overall, we analyzed why should incorporate adversarial training on real samples into GAN training and showed the proposed method can improve GAN training greatly.
>
> 2.’ In regarding robust optimization, I think [2] could cover a lot of practices considered in this paper. Some questions listed here are relevant to this point:
>     (a) The paper only considers to use FGSM in adversarial training part, but FGSM training on large epsilon usually leads to overfitting [2] on CIFAR-10. I wonder if the authors have tried PGD counterpart in their method.
> (b) It seems that the method uses both natural and adversarial examples in adversarial training, as in [3]. Instead, there is another (and perhaps more common) type of adversarial training [2] that uses only adversarial examples for the training. What happens if this training is applied to the method?’
>
> Response:
> (a)We have tried PGD but found no considerable improvement (<1 w.r.t FID) than FGSM but the computation overhead is relatively large, So we do not adopt this attack scheme. The computation overhead of our method is about 25%, but PGD attack needs several forward pass and backward pass. Here is comparison of average training time of one epoch measured on single RTX 2080ti.
>
> -------------------------------------------------------------------------------------------
> Setting              DCGAN   AS-DCGAN(ours)   SN-DCGAN   DCGAN-GP
> Training time   19.83s     26.40s                      24.50s           31.57s
> --------------------------------------------------------------------------------------------
>
> Perturbation level we use in the paper is small (1/255) and we do not observe overfitting.
> (b)We do extensive experiments that only performing training on adversarial samples and find that training become unstable and the loss collapses. Hence, we adopt performing adversarial training both on real samples and adversarial samples.

---

> ### Author Response · Authors · 2019-11-08
> **Response for reviewer#3 part 2**
>
> Thank you for your detailed and thoughtful review.  We will address your concerns one by one below.
>
> 3.’ - The method makes an additional parameter updates (Ep. 3 and 8) for adversarial training. Could this step make additional gain to the method by training more, i.e. perhaps it is a bit unfair to the vanilla training?’
>
> Response:
> Actually, we have did ablation study in Sec 4.1 and appendix A.2 with zero perturbation and Gaussian noise perturbation. Results show that performance of these two settings can not exceed the baseline. We can draw the conclusion that training twice at same data (zero perturbation) and training with random perturbation (Gaussian noise) can not work. Hence, the comparison is fair given the above experiment results.
>
> 4.’ - It would strengthen the claim if the paper could present whether the robustness of D is indeed increased, e.g. by comparing adversarial accuracies?’
>
> Response:
> We have compared robustness of standard discriminator and adversarially trained discriminator in appendix D. Results show that the accuracy of adversarially trained discriminator is much high than standard discriminator at the same perturbation level (1/255).
>
> 5.’ - Table 1: I feel there should be more discussion about why the reproduced values are so different compared to that of previously reported values, as they might confuse the readers to convince the claimed results.’
>
> Response:
> Hyper-parameters such as learning rate of our reproduced DCGAN are same as original paper but the network structure is slightly different. We use deconvolution with kernel size 4 but original paper used 5. Moreover, we train all the models for 200 epoch. Because training standard GAN on CelebA is unstable, we decrease learning rate of discriminator to 5e-5 to balance training as TTUR training strategy. We do not use other tricks to improve FID score. FID is calculated by official implementation [2].
>
> 6.’ - Eq. 7: The first term in the right hand side have to be E[max (log D(x - d))] instead of E[(log D(x - d))]?’
>
> Response:
> Thanks for pointing it! It should be E[min (log D(x - d))] because confidence of D w.r.t adversarial samples is smaller than that of benign samples. We will correct it in the revised pdf.
>
> [1] Liu, X., & Hsieh, C. J. (2019). Rob-gan: Generator, discriminator, and adversarial attacker. CVPR 2019.
> [2] https://github.com/mseitzer/pytorch-fid

---

### Official Review · AnonReviewer2 · 2019-10-18
**Official Blind Review #2**

**Rating:** 6

**Review:**

This paper presents an interesting idea based on introducing adversarial noise on real samples during GAN training. This novel approach may improve GAN training and have potentially large impact, but the paper in its current form is slightly below the standard of ICLR due to its lack of clarity.

While it is very interesting to apply adversarial noise in real data, this approach is not clearly motivated or explained. For example, at the beginning of page 2, why “As a consequence, training will become unstable when generated distribution approximates target distribution because the gradient given by non-robust discriminator around real samples contains more adversarial noise”? One may think that, on the contrary, the adversarial noise in generated samples would approximate that in real data when the generator distribution approximate the true data distribution. Similarly, after eq. 6, why “Consequently, gradient given by discriminator may vanish when discriminator becomes stronger than generator without capacity constrained.”? I would think the gradient would explode in this case given all the regularisation on gradients.

In addition, the term “non-robust discriminator” has been used several times in important places, but is not clearly defined. Properties used to justify the approach, such as “symmetric” and “balanced” need to be explained. For example, would it be possible to illustrate or measure the imbalance of discriminator?

The overhead of the algorithm needs to be stated more precisely. (after eq. 8) It is unclear to me that “backward propagation of Equation 5 with respect to x with negligible computation overhead”. Would this require back-prop through the entire discriminator, which can be very deep thus costly? It would be helpful to provide an estimation of runtime overhead supported by experiments.

In algorithm 1, why is it necessary to perform discriminator update twice? How about skipping step 4 (eq. 12)? I think this may better mirror the adversarial update for generated samples, which only involves updating generator parameters once.

In section 3.4, I am not sure it is similar to unrolled GAN when the perturbation is zero. Unrolled GAN required backprop through discriminator update, which I don’t think is the case here

Finally, the experimental results are confusing. In Table 1, why the reproduced results are already much better than those in the original papers? A solid baseline is necessary for any further assessment. Further more, since the DCGAN and ResNet baseline models did not use recent regularisation approaches such as spectral-norm, it is hard to assess whether the proposed method can work without those existing techniques. It would be helpful to use at least a SN-GAN baseline.

The analysis also need to be improved. It is hard to interpret the histograms and L1 norms in Figure 5 as related to “informative gradient” or “semantic information” as claimed in sectiion 4.4. Quantitative measurements such as correlation or mutual information may justify these claims.

Overall, I think the idea presented is certainly promising, but needs further development to be qualified for acceptance.

Nit:

The 1-line derivation of eq. 6 can be incorporated into the main text.

A transpose in eq.10 is missing.

Update:

I read the authors' rebuttal, which addressed many of my concerns and presented additional empirical results. The improvement over SN-GAN baseline is particularly convincing. I therefore believe the final camera-ready version can be a valuable contribution to the field, and would like to recommend accepting this paper.

**Experience Assessment:**

I have published one or two papers in this area.

**Review Assessment: Checking Correctness Of Derivations And Theory:**

I carefully checked the derivations and theory.

**Review Assessment: Checking Correctness Of Experiments:**

I carefully checked the experiments.

**Review Assessment: Thoroughness In Paper Reading:**

I read the paper thoroughly.

---

> ### Author Response · Authors · 2019-11-08
> **Response for reviewer#2 part 1**
>
> Thank you for your detailed and thoughtful review.  We will address your concerns one by one below.
>
> 1.’While it is very interesting to apply adversarial noise in real data, this approach is not clearly motivated or explained. For example, at the beginning of page 2, why “As a consequence, training will become unstable when generated distribution approximates target distribution because the gradient given by non-robust discriminator around real samples contains more adversarial noise”? One may think that, on the contrary, the adversarial noise in generated samples would approximate that in real data when the generator distribution approximate the true data distribution. Similarly, after eq. 6, why “Consequently, gradient given by discriminator may vanish when discriminator becomes stronger than generator without capacity constrained.”? I would think the gradient would explode in this case given all the regularisation on gradients.’
>
> Response:
> we will explain our motivation more clear and precise in the revised version. Our motivation is that, gradient that guides updates of generator contains adversarial noise which can be used to craft adversarial samples of discriminator, which can mislead generator and make training unstable intuitively. Thus, we introduce adversarial training on real samples into standard GAN training framework to make discriminator more robust, which can reduce adversarial noise and can be validated by the gradient visualization as Fig 1. Training become more stable with less adversarial noise and this can be validated by training cures. Gradient of robust classifier contains less adversarial noise has been investigated in literature about adversarial samples [1]. Although adversarial training on fake samples has been implemented in GAN training and robustness of discriminator around fake samples is relatively high, standard discriminator is still vulnerable to perturbation around real samples, which is validated by experiments. Hence, it is necessary to further incorporate adversarial training on real samples.
>
> 2.’In addition, the term “non-robust discriminator” has been used several times in important places, but is not clearly defined. Properties used to justify the approach, such as “symmetric” and “balanced” need to be explained. For example, would it be possible to illustrate or measure the imbalance of discriminator?’
>
> Response:
> Non-robust discriminator means that discriminator without additionally adversarial training on real samples is vulnerable to well-crafted imperceptible perturbation, which can be validated by experiments in appendix D. Given that adversarial training on fake samples has been used in standard GAN training framework but adversarial training on real samples does not exist, ‘symmetric’ means adversarial training both on fake samples and real samples. ‘balanced’ means the capabilities of discriminator and generator are comparative and can be measured by ratio of loss, which is shown in appendix A.1. We will clarify these important terms more clear in the revised pdf.
>
> 3.’The overhead of the algorithm needs to be stated more precisely. (after eq. 8) It is unclear to me that “backward propagation of Equation 5 with respect to x with negligible computation overhead”. Would this require back-prop through the entire discriminator, which can be very deep thus costly? It would be helpful to provide an estimation of runtime overhead supported by experiments.’
>
> Response:
> Gradient of input can be obtained with slight overhead because it is side result of calculating gradient of parameters of discriminator using error backward propagation. Performing training on adversarial samples takes additional one forward pass and one backward pass, which does need considerable computation. Overall, the additional computation overhead is about 25% relative to DCGAN baseline. Here is comparison of average training time of one epoch measured on single RTX 2080ti.
> -------------------------------------------------------------------------------------------
> Setting              DCGAN   AS-DCGAN(ours)   SN-DCGAN  DCGAN-GP
> Training time   19.83s     26.40s                      24.50s           31.57s
> --------------------------------------------------------------------------------------------
> Computation overhead of AS-DCGAN is comparable to spectral normalization on discriminator but much smaller than gradient penalty.

---

> ### Author Response · Authors · 2019-11-08
> **Response for reviewer#2 part 2**
>
> Thank you for your detailed and thoughtful review.  We will address your concerns one by one below.
>
> 4.’ In algorithm 1, why is it necessary to perform discriminator update twice? How about skipping step 4 (eq. 12)? I think this may better mirror the adversarial update for generated samples, which only involves updating generator parameters once.’
>
> Response:
> Actually, it is feasible to perform single update of discriminator by feeding both real data and adversarial samples of real data at the same forward pass , and training procedure does become more symmetric. However, it needs to calculate second order derivative when backward propagating error. We adopt updating discriminator twice just for the sake of taking advantage of gradient of real data as side result in the first update. These two procedures can provide comparable improvement.
>
> 5.’ In section 3.4, I am not sure it is similar to unrolled GAN when the perturbation is zero. Unrolled GAN required backprop through discriminator update, which I don’t think is the case here’
>
> Response:
> The proposed method is not completely same as unrolled GAN when perturbation is zero. Unrolled GAN tries to update discriminator many times on different samples to converge, which is just used as surrogate loss when training generator but parameter of discriminator remains unchanged after updating the generator. The proposed method with zero perturbation updates discriminator twice on same samples, in which the second update changes the parameter of discriminator. Hence, these two settings are somewhat similar but not completely equivalent.
>
> 6.’ Finally, the experimental results are confusing. In Table 1, why the reproduced results are already much better than those in the original papers? A solid baseline is necessary for any further assessment. Further more, since the DCGAN and ResNet baseline models did not use recent regularisation approaches such as spectral-norm, it is hard to assess whether the proposed method can work without those existing techniques. It would be helpful to use at least a SN-GAN baseline.’
>
> Response:
> Hyper-parameters such as learning rate of our reproduced DCGAN are same as original paper but the network structure is slightly different. We use deconvolution with kernel size 4 but original paper used 5. Moreover, we train all the models for 200 epoch. Because training standard GAN on CelebA is unstable, we decrease learning rate of discriminator to 5e-5 to balance training as TTUR training strategy. FID is calculated by widely adopted implementation [2]. In order to validate that the proposed method can work with wide adopted existing regularization techniques, we perform experiments with spectral normalization on discriminator. Results show that the proposed method can further improve performance of network with spectral normalization. Here are the FID scores.
> ---------------------------------------------------------
> Setting                           CIFAR10      CelebA
> SN-DCGAN                     30.47          19.68
> AS-SN-DCGAN(ours)    24.50          10.60
> ---------------------------------------------------------
>
> 7.’The analysis also need to be improved. It is hard to interpret the histograms and L1 norms in Figure 5 as related to “informative gradient” or “semantic information” as claimed in sectiion 4.4. Quantitative measurements such as correlation or mutual information may justify these claims.’
>
> Response:
> We will make analysis more clear in the revised pdf. Semantic information and informative gradient are shown by Fig 1 at the beginning of the paper. Gradient of adversarially trained discriminator is more sparse and contains semantic part such as profile of face, which indicates that adversarial noise is partly eliminated.
>
> Minor:
> 1. We incorporate 1-line derivation into main text.
> 2. Columns of the Jacobian matrix is equal to the size of \phi, so no transpose is missing?
>
> Thank you for your review!
>
> [1] Dimitris Tsipras, Shibani Santurkar, Logan Engstrom, Alexander Turner, and Aleksander Madry.Robustness may be at odds with accuracy.
> [2] https://github.com/mseitzer/pytorch-fid

---

### Official Review · AnonReviewer1 · 2019-10-22
**Official Blind Review #1**

**Rating:** 3

**Review:**

This paper tries to propose a new method to stabilize the training procedure of GAN. To this end, they use adversarial samples of real data to train the discriminator, and claim that it is helpful to reduce the adversarial noise contained in the gradient. Although training GAN with adversarial samples of discriminator is somewhat novel, the method and experiments are not convincing.
I do not recommend the acceptance based on the limited contribution of this paper. The following is a detailed evaluation.

1. The paper uses vague description such as “This approach can improve the robustness of discriminator and reduce adversarial noise contained in gradient” without convincing justification. Please give a formal description or notation of “adversarial noise contained in gradient”, and discuss how to remove the effect of “adversarial noise” in principle instead of extensively testing adversarial training of discriminator.

2. The experiment is not convincing and the improvement is not significant. The author running adversarial training on CIFAR10 dataset with FGSM and the perturbation is tested from 0.2/255 ~ 4/255. The performance (FID score) is a bit sensitive to the amount of perturbation level. Moreover, this The improvement over DCGAN is quite limited given previous works such as WGAN-GP. Combined together, the result is not convincing (it seems to be a heavy tuning result rather than a principled solution).

**Experience Assessment:**

I have read many papers in this area.

**Review Assessment: Checking Correctness Of Derivations And Theory:**

I assessed the sensibility of the derivations and theory.

**Review Assessment: Checking Correctness Of Experiments:**

I assessed the sensibility of the experiments.

**Review Assessment: Thoroughness In Paper Reading:**

I read the paper at least twice and used my best judgement in assessing the paper.

---

> ### Author Response · Authors · 2019-11-08
> **Response for reviewer#1**
>
> Thank you for your thoughtful and detailed review. We will address your concerns one by one below.
>
> 1 . ”The paper uses vague description such as “This approach can improve the robustness of discriminator and reduce adversarial noise contained in gradient” without convincing justification. Please give a formal description or notation of “adversarial noise contained in gradient”, and discuss how to remove the effect of “adversarial noise” in principle instead of extensively testing adversarial training of discriminator.”
>
> Response:
> We will make description of main claim more clear in the revised version. Adversarial noise means the component in gradient of discriminator used to update generated images, which can not improve the fidelity of generated images but can drastically change the prediction of the discriminator. Existence of adversarial noise is because the decision boundary of discriminator is not ideal ,ie, discriminator as a classifier realized by a neural network is vulnerable to small well-crafted perturbation, eg, perturbation in gradient direction, which is an universal property of neural networks [1]. Intuitively, adversarial noise as guide signal can mislead generator so as to make training unstable. To this end, we introduce adversarial training on real samples that does not exist in GAN training framework, which can make discriminator more robust and smooth the decision boundary so as to reduce adversarial noise. This can be validated by the gradient visualization shown as Fig 1. Gradient given by standard discriminator seems like noise but gradient given by adversarially trained discriminator contains less noise and more semantic information, eg, profile of face.
>
> 2. The experiment is not convincing and the improvement is not significant. The author running adversarial training on CIFAR10 dataset with FGSM and the perturbation is tested from 0.2/255 ~ 4/255. The performance (FID score) is a bit sensitive to the amount of perturbation level. Moreover, this The improvement over DCGAN is quite limited given previous works such as WGAN-GP. Combined together, the result is not convincing (it seems to be a heavy tuning result rather than a principled solution).
>
> Response:
> It is reasonable that FID score is sensitive to perturbation level because too small perturbation (0~0.1/255 ) can not improve the robustness of discriminator and too large perturbation (>5/255) can drastically degrade real samples so as to mislead discriminator, which is clarified in Sec 3.4 and 4.1 . We validated that best performance improvement can be achieved with default setting of perturbation level (1/255 on image generation tasks) on CIFAR10, CelebA, LSUN with DCGAN and ResNet architecture. FID score is improved about 50% on CelebA, and 35% on LSUN. This is a significant improvement, which can not be realized by parameter tuning. We do not use other tricks to improve FID score. Suitability of default perturbation level is quite well and it is not required for heavy searching of perturbation level when applying the proposed method on other datasets with different network architectures. Extensive experiments validated that improvement is general and of principle but does not depend on heavy hyper-parameter tuning. Moreover, the proposed method is much efficient than gradient penalty.
> Here is comparison of average training time of one epoch measured on single RTX 2080ti.
> ----------------------------------------------------------------------------------------
> Setting              DCGAN   AS-DCGAN(ours)   SN-DCGAN  DCGAN-GP
> Training time   19.83s     26.40s                      24.50s          31.57s
> ----------------------------------------------------------------------------------------
>
> Thank you for your review!
>
> [1] Szegedy C, Zaremba W, Sutskever I, et al. Intriguing properties of neural networks[J]. arXiv preprint arXiv:1312.6199, 2013.

---

### Author Response · Authors · 2019-11-08
**PDF has been revised**

Dear reviewers,
We have uploaded the revised pdf. The main changes are as follows.
-emphasize our motivation and clarify it more clear.
-make vague description more precise and quantitative.
-correct E[(log D(x - \delta))] by E[min (log D(x - \delta))].
-incorporate short proof into main text.
-add computation overhead analysis.
 Thanks for your review!

---

### Decision · Program_Chairs · 2019-12-19

**Decision:**

Reject

**Comment:**

This paper proposes incorporating adversarial training on real images to improve the stability of GAN training. The key idea relies on the observation that GAN training already implicitly does a form of adversarial training on the generated images and so this work proposes adding adversarial training on real images as well. In practice, adversarial training on real images is performed using FGSM and experiments are conducted on CelebA, CiFAR10, and LSUN reporting using standard generative metrics like FID.

Initially all reviewers were in agreement that this work should not be accepted. However, in response to the discussion with the authors Reviewer 2 updated their score from weak reject to weak accept. The other reviewers recommendation remained unchanged. The core concerns of reviewers 3 and 1 is limited technical contribution and unconvincing experimental evidence. In particular, concerns were raised about the overlap with [1] from CVPR 2019. The authors argue that their work is different due to the focus on the unsupervised setting, however, this application distinction is minor and doesn’t result in any major algorithmic changes. With respect to experiments, the authors do provide performance across multiple datasets and architectures which is encouraging, however, to distinguish this work it would have been helpful to provide further study and analysis into the aspects unique to this work -- such as the settings and type of adversarial attack (as mentioned by R3) and stability across GAN variants.

After considering all reviewer and author comments, the AC does not recommend this work for publication in its current form and recommends the authors consider both additional experiments and text description to clarify and solidify their contributions over prior work.

[1] Liu, X., & Hsieh, C. J. (2019). Rob-gan: Generator, discriminator, and adversarial attacker. CVPR 2019.